# Quality indicators for the primary prevention of cardiovascular disease in primary care: A systematic review

Kiran Bam[1], Muideen T. Olaiya[1], Dominique A. Cadilhac[1,2], Julie Redfern[3,4,5], Mark R. Nelson[6,7], Lauren M. Sanders[8,9], Vijaya Sundararajan[10], Nadine E. Andrew[11,12], Lisa Murphy[13], Monique F. Kilkenny[1,2]*

1 Stroke and Ageing Research, Department of Medicine, School of Clinical Sciences at Monash Health, Monash University, Clayton, Victoria, Australia, 2 Stroke Theme, The Florey Institute of Neuroscience and Mental Health, University of Melbourne, Heidelberg, Victoria, Australia, 3 School of Health Sciences, Faculty of Medicine and Health, University of Sydney, Sydney, New South Wales, Australia, 4 The George Institute for Global Health, Sydney, New South Wales, Australia, 5 Institute for Evidence-Based Healthcare, Bond University, Robina, Queensland, Australia, 6 Menzies Institute for Medical Research, University of Tasmania, Hobart, Tasmania, Australia, 7 School of Public Health and Preventive Medicine, Monash University, Prahran, Victoria, Australia, 8 Department of Neurosciences, St Vincent's Hospital Melbourne, Fitzroy, Victoria, Australia, 9 Department of Medicine, University of Melbourne, Heidelberg, Victoria, Australia, 10 Department of Medicine, St Vincent's Hospital, Melbourne Medical School, University of Melbourne, Heidelberg, Victoria, Australia, 11 Peninsula Clinical School, Central Clinical School, Monash University, Frankston, Victoria, Australia, 12 National Centre for Healthy Ageing, Monash University, Frankston, Victoria, Australia, 13 Stroke Foundation, Melbourne, Victoria, Australia

* Monique.Kilkenny@monash.edu

**Data Availability Statement:** All relevant data are within the manuscript and its Supporting information files.

## Abstract

### Background

Primary care is usually the entry point for preventing cardiovascular disease (CVD). Quality indicators can be used to assess and monitor the quality of care provided in a primary care setting. In this systematic review, we aimed to identify, summarise, and assess the methodological quality of indicators reported in the articles for the primary prevention of CVD in primary care.

### Methods

We searched Ovid MEDLINE, Ovid EMBASE, CINAHL Plus, SCOPUS, and grey literature for articles containing quality indicators published in English language. Quality indicators were categorised using the Donabedian framework: Structure (*organisation of care*), Process (*assessment of metabolic risk factors*, *global risk assessment*, *lifestyle management*, *prescription of medications*, *risk communication/advice*, *referral*), and Outcome (*attainment of risk factor targets*). Articles were reviewed by two authors, using the Appraisal of Indicators through Research and Evaluation (AIRE) instrument, where a score of ≥50% for each domain indicated strong methodological quality (*e.g., stakeholder involvement*).

**Funding:** Kiran Bam received the Monash International Tuition Scholarship and Monash Graduate Scholarship support from Monash University, Melbourne, Victoria, Australia. Dominique A Cadilhac and Monique F Kilkenny received research fellowship support from the National Health and Medical Research Council of Australia (DAC: 1154273, MFK 1141848). Monique F Kilkenny received fellowship support from the National Heart Foundation of Australia (MFK 105737).

**Competing interests:** Dominique A Cadilhac received restricted grants from Boehringer Ingelheim, Moleac, Amazon Web Services, Nicolab, Philips and Medtronic; outside the submitted work. This does not alter our adherence to PLOS ONE policies on sharing data and materials.

## Results

We identified 282 articles for full-text review; 57 articles were included for extraction. A total of 726 (681 unique) quality indicators were extracted. Three out of four (76%) were process indicators (56 articles), followed by 15% outcome indicators (40 articles), and 9% structure indicators (12 articles). One-third of process indicators were related to the assessment of metabolic risk factors (222/726 indicators, 41 articles), followed by lifestyle management (153/726 indicators, 39 articles), prescription of medications (122/726 indicators, 37 articles), and global risk assessment (27/726, 14 articles). Few indicators were related to risk communication/advice (20/726 indicators, 7 articles) and referral (9/726 indicators, 6 articles). Only 26/57 (46%) articles were found to have strong methodological quality.

## Conclusion

We summarised and appraised the methodological quality of indicators for the primary prevention of CVD. The next step requires prioritising a minimum set of quality indicators to encourage standardised collection and monitoring across countries.

## Introduction

Cardiovascular diseases (CVD), including heart disease and stroke, remain the leading global cause of death and disability, despite the availability of proven effective prevention interventions [1]. Of the estimated 7.9 million deaths worldwide in 2019, 85% were due to heart attack and stroke [2]. Further, the treatment costs of CVD are escalating globally [3]. Given the growing burden of CVD and its increasing treatment costs, effective prevention strategies are needed to prevent first-ever cardiovascular events (i.e. primary prevention).

Effective implementation of primary care services, e.g., risk factor assessment followed by appropriate lifestyle and pharmacotherapy interventions, is essential for preventing CVD [4]. Primary care is usually the entry point for prevention of CVD, and coordination for the ongoing management of risk factors, e.g., hypertension, diabetes [5]. Therefore, it is imperative that primary care services are routinely monitored to ensure optimal care. Similarly, routine monitoring of primary care services has become increasingly important to patients, clinicians, healthcare organisations and policymakers to measure quality of care [6]. There is an increasing pressure for primary care providers, especially, to ensure the funds and efforts spent are meeting clinical guideline recommendations [7].

Periodic monitoring and feedback on the services provided is needed to improve the quality of primary care [8]. Quality indicators can be used by primary care providers to quantitatively and qualitatively assess and monitor the quality of care provided in primary care [9]. These indicators are measurable tools for evaluating performance of primary care services concordant with evidence-based guidelines [7]. Moreover, they are also important for informing strategies (e.g., a quality improvement plan) for achieving optimal care and managing risk factors [10, 11].

An ideal set of quality indicators, as per country-specific clinical guidelines, should include a numerator, denominator, performance or threshold standard, and frequency of collection [4], while also being feasible for implementation with realistic effort, cost, and time [12]. Quality indicators usually follow the format of *if* (specific clinical scenario or condition), *then* (corresponding clinical action) for a particular condition (e.g. to prevent CVD or stroke) [10]. A

simplified commonly used example of a quality indicator for primary prevention of CVD could be "*If* a patient has high blood pressure, *then* blood pressure lowering medications should be prescribed and recorded".

There have been efforts in the development of the quality indicators for the prevention of CVD for specific settings [13, 14]. Despite these efforts, comprehensive reviews that summarise quality indicators across disease conditions or risk factors for primary prevention of CVD in primary care are lacking. Therefore, in this systematic review, we aimed to identify and summarise the quality indicators used globally and assess the methodological quality of these indicator sets that are being used to manage risk factors for the primary prevention of CVD.

## Methods

A systematic review was undertaken according to the Preferred Reporting Items for Systematic reviews and Meta-Analyses (PRISMA) 2020 guidelines [15]. PRISMA comprises 27-point checklist for the reporting of a systematic review (S1 File). The review was registered on the international prospective register of systematic reviews (PROSPERO ID number: CRD42022359131).

### Eligibility criteria

We developed a broader eligibility criterion to include all potential articles with sufficient details on the quality indicators for primary prevention of CVD. All study designs, except case report, case series, letter to editor, protocol, conference abstracts, were included. We included full-text articles published in English language with sufficient details on the definition of quality indicators. Further details on eligibility criteria are described according to Population, Interventions, Comparator and Outcome (PICO) elements:

- **Population:** People without CVD being assessed, managed or treated in the primary care setting for primary prevention of CVD.

- **Interventions:** Development, validation, implementation, and evaluation of quality indicators targeting the management of risk factors for CVD.

- **Comparator:** Condition, type of quality indicator, themes, domains of primary care services.

- **Outcome:** Quality indicators in for primary prevention of CVD.

### Search strategy

We searched Ovid MEDLINE, Ovid EMBASE, CINAHL Plus, SCOPUS, and grey literature from January 2012 to August 2022. To ensure relevant new articles were included, citation tracking of eligible articles was undertaken in October 2023. The search strategy was developed in consultation with a research librarian, to identify appropriate search terms. Inputs were also obtained from an advisory group comprising topic experts, i.e. a general practitioner, epidemiologists, implementation researchers, and quality improvement experts. The systematic review required a carefully designed search strategy, due to the diverse terminologies and varied definitions used for quality indicators for CVD across varying settings. The search terms used were "cardiovascular disease" or "stroke" AND "risk factors" AND "quality indicators" AND "primary care" AND "prevention". Boolean operators such as AND, and OR were used to combine these search terms. The search terms were identified based on the initial scoping

searches of the literature in Google Scholar and refined based on five relevant articles. Truncation (e.g., cardiovasc*, cerebrovasc*, stroke*) and Wildcards (e.g., cardiovasc$, cerebrovasc$) were used to broaden results and capture variations of words. We searched each database using a range of Medical Subject Headings (MeSH) terms to find all potentially relevant articles. The summary of the different databases and full search strategy is documented in S1 Table. Below is an example of the search terms used in CINAHL Plus:

(Cardiovasc* adj1 disease* OR heart adj1 disease* OR Stroke* OR Transient isch*or Haemorrhag* adj1 stroke OR hemorrhag* adj1 stroke OR TIA OR myocardial infarction OR Cerebrovasc* event* OR Cerebrovasc* disease* OR Cerebrovasc* accident* OR cardiac failure) AND (Risk* OR Hypertension OR High adj2 pressure OR Dyslipi* OR Hyperlipi* OR atrial adj1 fibrillation OR Smoking OR Diabetes OR diabetes mellitus OR Alcohol OR Physical adj2 activity OR Obesity OR Mental health OR mental disorder* OR depress* OR anxiety OR psychiat* OR well-being OR quality of life OR self esteem OR self perception) AND (Quality adj1 care* OR Quality assessment OR Quality indicator* OR Quality assurance OR medication* OR medicine* Quality improvement OR Quality tool* OR Quality monitor* OR Quality metric OR Quality criter* OR Performance indicator OR Process adj2 care OR Performance measure OR Benchmark OR benchmarking OR Outcome measure OR Outcome indicator* OR Consensus OR delphi) AND (Primary adj2 care OR General adj1 practi* OR general practice OR general practitioners OR Clinical practice OR Family practice OR primary adj2 physician OR healthcare delivery) AND (primary adj2 prevention OR secondary adj2 prevention OR prevention OR control).

We searched grey literature in the form of unpublished empirical research papers and databases on quality indicators primarily from countries with similar, well-established primary care systems, such as the United Kingdom (UK), Canada and Australia. Grey literature was searched using google scholar and websites of respective government or non-government organisations (e.g., National Institute for Health and Clinical Excellence Australian Institute of Health and Welfare). The list of organisations researched are documented in S2 Table.

### Article screening

Articles obtained from the literature search were managed in EndNote version 19. The duplicates were identified and removed through EndNote using "find duplicates" feature. Manual review was also performed to check for any remaining duplicates, and then imported into Covidence for screening and data extraction [16]. No other automated tools were used to eliminate the articles. KB, MFK and MTO screened the titles and abstracts of articles for eligibility. At this stage, we excluded articles that were not related to quality indicators for cardiovascular disease or stroke based on title and abstract. Any conflicts between reviewers were resolved together by KB, MFK and MTO. To facilitate full-text screening, a review checklist (S3 Table) was developed to check against eligibility criteria. Full-text screening was done by KB, MFK and MTO. Disagreements between any two authors were adjudicated by DAC. At this stage, the primary reason for exclusion was recorded and summarised in the PRISMA flowchart (Fig 1).

### Data extraction and synthesis

The excel file template used for data extraction was initially developed by KB, MFK, MTO, and DAC, with inputs from the advisory group (other co-authors), and modified as the extraction progressed. The final data extraction table for study characteristics included information for author, year of publication, country, study design/methodology, and study period (S4 Table). Other important information collected on summary of quality indicators include condition,

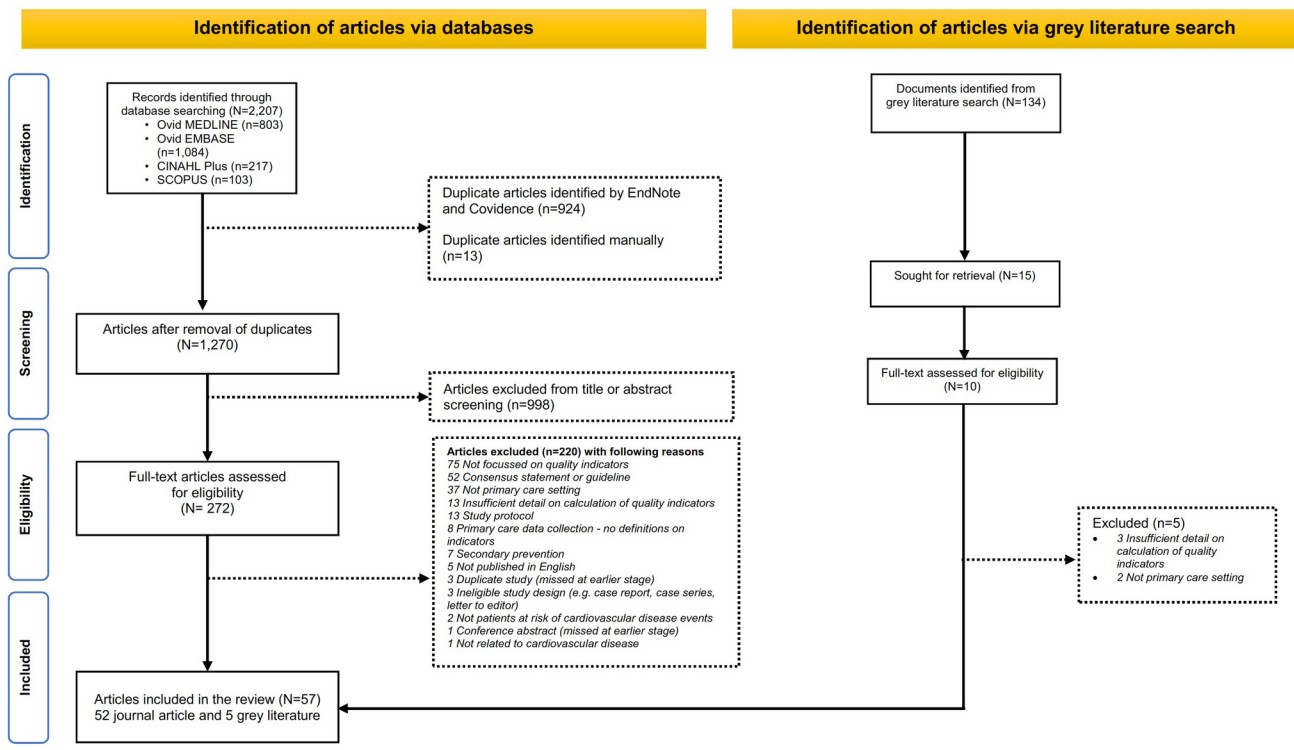

**Fig 1. PRISMA 2020 flowchart for identification and inclusion of articles.**

domains, theme, type of indicator, age group (years), numerator, denominator, and the frequency of collection (S5 Table). Missing information was indicated as not specified. Data extraction was undertaken by KB, reviewed by MTO, and any disagreements were adjudicated by MFK or DAC.

Details about the quality indicators including numerator, denominator, age group (in years) and frequency of collection were extracted and summarised in tabular format (S5 Table). For further data synthesis, themes and conditions were identified based on literature [14, 17–21], pre-tested across the initial set of the articles. For categorisation of domains and type of indicator, a theory informed conceptual framework (Fig 2) was developed from existing literature [14, 18, 20–23], and the Donabedian's multidimensional framework [24, 25] of structure, process and outcome was used. This framework facilitated a comprehensive summary and evaluation of care quality across domains of primary care services [26], as described below.

a. **Structure indicators** include infrastructure or human resources-related indicators that designate the conditions under which primary care is provided (*organisation of care*);

b. **Process indicators** include the actions and processes of clinical care delivered by primary care physician, such as screening, ordering diagnostic tests, prescribing treatment i.e. medications, or monitoring (*assessment of metabolic risk factors*, *global risk assessment*, *lifestyle management*, *prescription of medications*, *risk communication/advice*, *or referral*); and

c. **Outcome indicators** reflect changes in individuals resulting from the intervention provided at the primary care (*attainment of risk factor targets*).

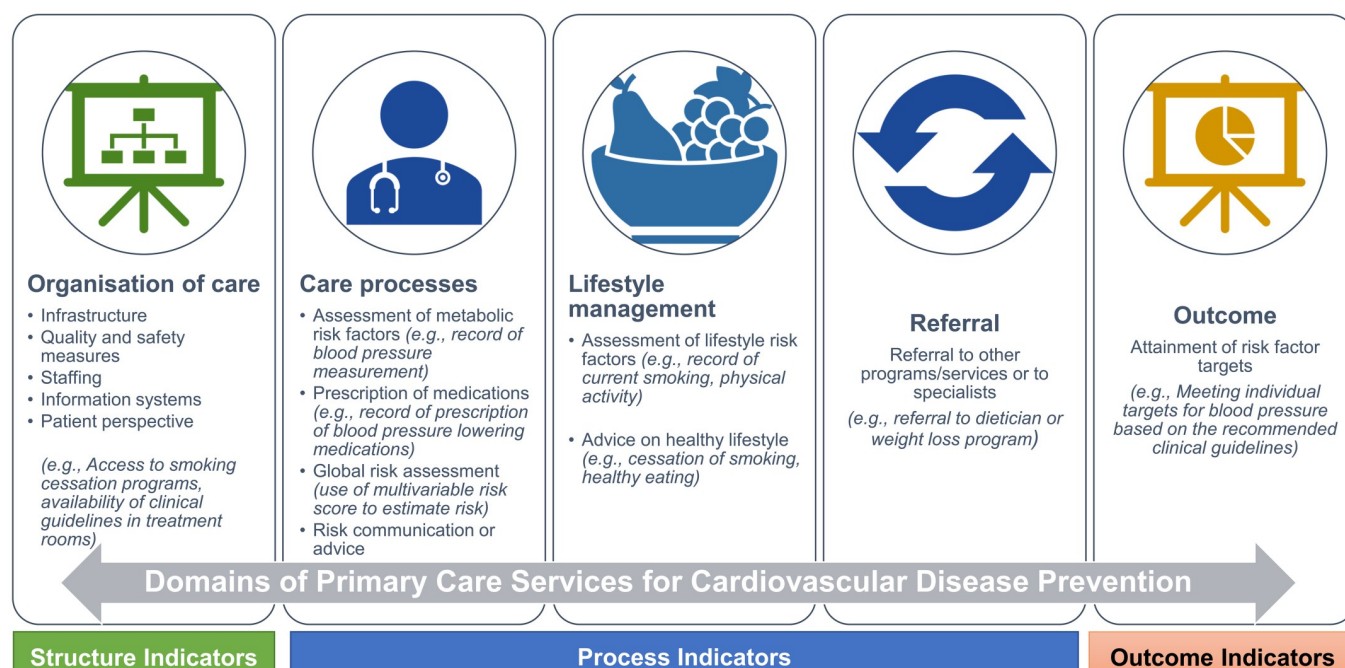

**Fig 2. Conceptual framework indicating the domains of primary care services for cardiovascular disease prevention.**

## Methodological quality appraisal of the indicators in the articles (including grey literature)

Quality appraisal of the indicators reported in the articles and their development process was undertaken using the Appraisal of Indicators through Research and Evaluation (AIRE) instrument [27, 28] (S6 Table). The AIRE instrument is a validated and reliable tool to appraise the methodological quality of indicators [29]. The AIRE instrument is used by researchers, evaluators, and policymakers to enable the standardised assessment of the robustness and relevance of indicators [30]. The tool was developed from the Appraisal of Guidelines through Research and Evaluation (AGREE) instrument [31], a commonly used instrument to assess the methodological rigor. The AIRE has been used previously in systematic reviews in various settings [32–36]. The AIRE instrument comprises 20 items across four domains: 1) purpose, relevance and organisational context; 2) stakeholder involvement; 3) scientific evidence; and 4) additional evidence, formulation, and usage. Each domain of the AIRE instrument has a Likert scale score ranging from 1 to 4, with 1 indicating strongly disagree (confident that the criterion has not been fulfilled or no information was available); 2 or 3 indicating disagree/agree (unsure whether the criterion has been fulfilled); and 4 indicating strongly agree (confident that the criterion has been fulfilled). The sum of the scores for each domain was calculated and standardised as a percentage, with increasing percentage denoting a higher score and strong methodological quality. The AIRE instrument was used to review each article as a single quality indicator set instead of individual quality indicator reported in the articles. This method was applied because the articles comprised overall information about the evidence and development process for the overall quality indicator set rather than each separate indicator. Articles with a score of ≥50% in each domain were considered to have strong methodological quality for development of quality indicators. This criterion was based on previous evaluations

that have been undertaken for the appraisal of quality indicators in other systematic reviews [32, 33]. A template created as an excel file was used to appraise the included articles. KB appraised the methodological quality of articles with the AIRE instrument. A random 10% sample of articles was audited by co-authors (MFK and MTO) to ensure reliability of the assigned quality ratings.

### Data analysis

Descriptive statistics (frequencies and percentages) were used to summarise results using Microsoft Excel's advanced analysis features, particularly by pivot tables. Data were presented in tabular format with narrative description given the nature of the review. Figures, where applicable, were developed to visualise the flow diagram, methodological quality (e.g., heat map), and distribution of quality indicators using Microsoft Excel or Microsoft PowerPoint graph features.

## Results

### Search results and characteristics

A total of 2,341 articles were identified, including 2,207 through database searches, and 134 documents in grey literature. After removing duplicates and excluding articles from title or abstract screening, we identified 282 articles for full-text screening (including 272 from journal article database searches and 10 from the grey literature). Each of the excluded articles, along with the reason for its exclusion, is presented in S7 Table.

Following full-text screening, 57 articles (including five from grey literature) met the eligibility criteria (Fig 1). A total of 726 quality indicators were extracted (S8 Table) from the 57 articles (681 unique quality indicators by disease conditions; S9 Table), across 50 themes, categorised for the domains for the primary prevention of CVD (Table 1). Most of these articles originated from high-income countries (89.5%), including 15 from the UK (26.3%), 11 from Australia (19.3%), eight from Canada (14.0%) and six from the United States (10.5%; Table 2).

In four articles (7.0%) [13, 20, 21, 37], a combination of systematic or literature reviews and Delphi or modified Delphi method were used to develop quality indicators (S10 Table). Consensus-based approach were used to develop quality indicators in only five articles (9.0%) [13, 20, 21, 37, 38]. For process indicators, 30 articles (52.6%) incorporated benchmarking, and incentives to encourage the recording of these quality indicators reported in five articles (8.8%) [39–43]. In the UK, quality indicators aimed at fostering the systematic recording of the care provided through financial incentives were reported in four articles (7.0%) [40–43]. Similarly, the implementation of incentivisation for 10 quality indicators to promote the systematic recording of care provided to patients was reported in an Australian article [39].

### Type of quality indicators as per Donabedian framework

Process indicators were reported in the majority (56/57) of the articles (S11 Table). Only six articles [13, 20, 44–47], comprised a combination of structure, process, and outcome indicators.

**Structure indicators.** Quality indicators on the organisation of care were reported in only 12 articles, contributing to 65/726 of all quality indicators (9.0%). These quality indicators were used to evaluate the organisational and management systems, encompassing aspects such as infrastructure reported in five articles [13, 45, 47–49], comprising 11/726 indicators, information systems (31/726 indicators) reported in four articles [13, 20, 46, 50], staffing (9/726 indicators) in four articles [13, 20, 46, 50], safety of patients (7/726 indicators) reported in

**Table 1. Summary of indicators by domains of primary care services for cardiovascular disease prevention and themes.**

| Types/Domains | Summary of indicators by themes |
|---|---|
| **Structure indicators** | |
| Organisation of care [13, 20, 37, 44–52] | Service delivery, patient perspective, quality and safety, information, continued medical education or training, and multi-disciplinary team |
| **Process indicators** | |
| Assessment of metabolic risk factors [13, 20, 21, 37–44, 46–70, 72–76] | Blood pressure, blood lipid, blood glucose, atrial fibrillation monitoring, kidney function test, depression or mental illness, and quality of life assessment |
| Global risk assessment [13, 37, 39, 49, 50, 56, 66, 74, 77, 78] | Framingham risk score, World Health Organization (WHO)/International Society of Hypertension (ISH) risk scores, and assessment of multiple risk factors |
| Assessment of lifestyle factors [13, 20, 21, 39, 40, 42, 44, 46–51, 53–57, 59–64, 66, 70, 73, 76, 79–81] | Smoking status, physical activity or exercise, alcohol consumption status, stress intensity level, weight/height or waist circumference, and fruit and vegetables intake |
| Advice on healthy lifestyle [20, 21, 39, 42–44, 46–48, 50, 55, 60–62, 64, 65, 70, 71, 73, 75, 82–85] | Healthy eating, physical activity or exercise, low-risk alcohol drinking, weight management, smoking/tobacco cessation, and advice/uptake of immunisation or vaccination |
| Prescription of medications [13, 20, 21, 37, 38, 41, 43, 44, 47–55, 59, 60, 62–64, 69, 71, 72, 74, 75, 77–81, 84–88] | Blood pressure-lowering medications, lipid-lowering medications, glucose-lowering medications, anticoagulant medications, antiplatelet medications, aspirin use, inappropriate prescription of medicines, adherence to medications, smoking cessation medication, and management of multiple risk factors |
| Risk communication/advice [13, 20, 44–46, 50, 68] | Care plan, self-management plan, pharmaceutical opinion, and information leaflet |
| Referral [40, 47, 50, 60, 62, 70] | Smoking cessation, dietician or weight loss program, retina examination, and computed tomography or nuclear magnetic resonance |
| **Outcome indicators** | |
| Attainment of risk factor targets [13, 20, 21, 41, 42, 44–47, 51, 53–56, 59, 61, 63–66, 68–71, 73–85, 87, 89, 90] | Blood pressure, serum lipids, body weight, smoking cessation, alcohol, and physical activity. |

three articles [37, 51, 52], capacity building (e.g., training, continued medical education; 7/726 indicators) reported in two articles [20, 46], and patient perspectives on care received (2/726 indicators) reported by only two articles [13, 37]. An indicator for the availability of a multidisciplinary team was only reported in a single article [13]. There was no single comprehensive set of quality indicators that comprised all aspects of the structure indicators.

**Process indicators.** Quality indicators related to processes of care including assessment of metabolic risk factors, global risk assessment, lifestyle management, prescription of medications, risk communication/advice, or referral were reported in 56 articles (98.2%), comprising 553/726 of quality indicators.

*Assessment of metabolic risk factors.* Quality indicators focussing on assessment of metabolic risk factors were reported in 41 articles, comprising 222/726 (30.6%) indicators. Most frequently reported quality indicators were blood pressure monitoring (55/726 indicators) [20, 21, 39, 41–44, 46–51, 53–67], and blood glucose level monitoring (59/726 indicators) [13, 20, 21, 39–44, 46–50, 53–57, 61–69]; both of them reported in 28 articles, followed by serum lipids assessment (34/726 indicators) reported in 23 articles [20, 21, 42, 43, 46, 48, 50, 53, 54, 57, 59–71]. Quality indicators for the assessment of metabolic risk factors also included screening of

**Table 2. Characteristics of included articles.**

| Characteristics | N = 57<br>n (%) |
|---|---|
| **Region/countries** | |
| Asia | 3 (5.4) |
| Europe | 13 (23.2) |
| Australia | 11 (19.3) |
| Canada | 8 (14.0) |
| United Kingdom | 15 (26.3) |
| United States | 6 (10.5) |
| South America | 1 (1.8) |
| **Type of study** | |
| *Implementation and evaluation* | |
| Cross sectional study | 19 (33.3) |
| Cohort study | 13 (22.8) |
| Randomised controlled trial | 9 (15.8) |
| Quality improvement study | 5 (8.8) |
| Mixed methods | 1(1.8) |
| Pre-post study | 1 (1.8) |
| *Development and validation* | |
| Grey literature | 5 (8.8) |
| Review and (Modified) Delphi method | 4 (7.0) |

atrial fibrillation (32/726 indicators) [21, 37, 38, 47, 50–52, 58, 64, 72–74], and kidney function test (23/726 indicators) [41, 43, 47–50, 55, 60, 62–64, 66]; both of them reported in 12 articles. Least reported indicators were assessment of emotional well-being (including depression or mental health illness) reported in only four articles [37, 43, 50, 63], and history of CVD reported in two articles [54, 75]. Similarly, quality of life assessment reported in two articles [37, 76]. Evaluation of physical function or assessment of cognitive function was reported in a single article [37].

*Global risk assessment*. Quality indicators for global risk assessment were reported in 14 articles, contributing 27/726; 3.7% of all indicators. Among these, the Framingham CVD risk score was reported in six articles [20, 44, 56, 64, 66, 77]. A single article reported use of color-coded charts [78], and the World Health Organization (WHO)/International Society of Hypertension (ISH) risk scores [54]. The categorisation of risk based on the assessment of multiple risk factors (18/726 indicators) was reported in 10 articles [13, 37, 39, 49, 50, 56, 66, 74, 77, 78]. However, there were no indicators for global risk assessment specifically reported for individuals with existing risk factors such as kidney disease, dyslipidaemia, transient ischemic attack, and heart failure.

*Lifestyle management indicators*. Quality indicators related to lifestyle management were reported in 39 articles, comprising 153/726; 21.1% of all quality indicators. Assessment of lifestyle factors comprised 97/726; 13.4% of all indicators were reported in 31 articles. However, advice on healthy lifestyle was only reported in 24 articles comprising 56/726 indicators; 7.7% of all indicators (Table 3).

For the assessment of lifestyle risk factors, most commonly reported indicator was smoking status (36/726 indicators) in 30 articles [13, 20, 21, 37, 39, 40, 42, 44, 46–50, 53, 54, 56, 57, 59, 61–63, 66, 70, 73, 74, 76, 77, 79–81], followed by measurement of body mass index (31/726 indicators) reported in 21 articles [20, 21, 39, 40, 42, 44, 46, 50, 51, 53, 55–57, 61, 63, 64, 66, 74, 76, 79, 81], assessment of physical activity (11/726 indicators) reported in 10 articles [20, 37,

**Table 3. Summary of indicators for primary prevention of cardiovascular diseases by type of indicators and domains.**

| Types/Domains | Number of indicators (N = 726) n (%) | Number of articles (N = 57) n (%) |
|---|---|---|
| **Structure indicators** | | |
| Organisation of care[a] | 65 (9.0) | 12 (21.1) |
| **Process indicators** | 553 (76.2) | 56 (98.2) |
| Assessment of metabolic risk factors | 222 (30.6) | 41 (71.9) |
| Global risk assessment[b] | 27 (3.7) | 14 (24.6) |
| Lifestyle management | 153 (21.1) | 39 (68.4) |
| *Assessment of lifestyle factors* | *97 (13.4)* | *31 (54.4)* |
| *Advice on healthy lifestyle* | *56 (7.7)* | *24 (42.1)* |
| Prescriptions of medications | 122 (16.8) | 37 (64.9) |
| Risk communication/advice | 20 (2.8) | 7 (12.3) |
| Referral | 9 (1.2) | 6 (10.5) |
| **Outcome indicators** | | |
| Attainment of risk factor targets | 108 (14.9) | 40 (70.2) |

[a]Organisation of care includes infrastructure, quality and safety measures, staffing, information systems, and the patient perspective within the system;
[b]Global risk assessment includes assessment of absolute cardiovascular risk factors or multiple risk factors.

44, 46, 50, 56, 57, 61, 76, 81], alcohol consumption status (10/726 indicators) reported in 9 articles [37, 39, 44, 49, 50, 56, 57, 66, 81], and measurement of waist circumference (12/726 indicators) reported in 8 articles [20, 48, 56, 57, 60, 62, 64, 68]. The least frequently reported indicators were assessment of fruit and vegetable intake (2/726 indicators) [56, 76], alongside assessment of diet intake status [44, 56], and salt consumption [44, 76], reported in just two articles. Both sleep apnoea [37], and stress intensity level [44], were reported in one article each. Notably, there was lack of quality indicators for assessing lifestyle risk factors in individuals with atrial fibrillation or kidney disease.

For advice on a healthy lifestyle, the most commonly reported quality indicators were on smoking cessation outlined in 19 articles (21/726 indicators) [20, 21, 40, 44, 46–48, 60–62, 64, 70, 71, 75, 78, 80, 82–84], followed by counselling or uptake of influenza or pneumococcal immunisation or vaccination (11/726 indicators) reported in 10 articles [39, 42–44, 47, 55, 65, 70, 73, 85], physical activity promotion (9/726 indicators) reported in six articles [20, 44, 46, 50, 61, 64], advice on healthy eating (7/726 indicators) reported in five articles, [20, 44, 46, 61, 64], advice on low-risk alcohol drinking reported in four articles (4/726) indicators [20, 44, 46, 64], and advice on weight control (4/726) reported in four articles [20, 44, 60, 62]. Least reported indicators were on advice on salt intake reduction [64], and lifestyle changes to reduce stress [44]; both reported in one article each. Specifically, quality indicators for lifestyle advice targeting individuals with atrial fibrillation, kidney disease, mental health, dyslipidaemia or transient ischaemic attack were lacking (Table 4).

*Prescription of medications.* Quality indicators on prescriptions of medications were reported in 37 articles comprising 122/726; 16.8% of all indicators. Most commonly reported indicators were blood pressure lowering medications (31/726 indicators) reported in 23 articles [20, 21, 41, 43, 44, 47, 48, 50, 51, 54, 55, 59, 60, 62, 64, 74, 77, 78, 81, 86–88], lipid-lowering medications (32/726 indicators) reported in 21 articles [13, 20, 21, 43, 44, 47–50, 53, 54, 59, 60, 62, 69, 75, 77–79, 84, 87, 88], anticoagulant medications (13/726 indicators) reported in 10

**Table 4. Quality indicators for primary prevention of cardiovascular diseases by existing risk factors or conditions.**

| Types/Domains | Indicators for people with no existing risk factor[a] (N = 116) | Indicators for existing risk factors or conditions, n (%) | | | | | | | | | | |
|---|---|---|---|---|---|---|---|---|---|---|---|---|
| | | At risk of CVD[b] N = 184 | Diabetes N = 149 | Hypertension N = 119 | Atrial fibrillation N = 64 | Kidney disease N = 25 | Mental health N = 25 | Dyslipidaemia N = 19 | TIA N = 17 | Smoking N = 17 | Heart failure N = 9 | Others[c] N = 11 |
| **Structure indicators** | | | | | | | | | | | | |
| Organisation of care | 33 (28.4)[d] | 5 (2.7) | 5 (3.4) | 5 (4.2) | 9 (14.1) | 2 (8.0) | - | - | - | 4 (23.5) | 1 (11.1) | 2 (18.2) |
| **Process indicators** | | | | | | | | | | | | |
| Assessment of metabolic risk factors | 37 (31.9) | 30 (16.3) | 52 (34.9) | 45 (37.8) | 32 (50.0) | 13 (52.0) | 12 (48.0) | 5 (26.3) | 4 (23.5) | - | 1 (11.1) | 1 (9.1) |
| Global risk assessment | 10 (8.6) | 6 (3.3) | 3 (2.0) | 7 (5.9) | 1 (1.6) | - | 2 (8.0) | - | - | - | - | - |
| Assessment of lifestyle risk factors | 23 (19.8) | 41 (22.3) | 8 (5.4) | 17 (14.3) | - | - | 6 (24.0) | 2 (10.5) | 1 (5.9) | 1 (5.9) | 1 (11.1) | - |
| Advice on healthy lifestyle | 6 (5.2) | 25 (13.6) | 9 (6.0) | 5 (4.2) | - | - | - | - | 1 (5.9) | 6 (35.3) | 2 (22.2) | 4 (36.4) |
| Prescription of medications | - | 30 (16.3) | 26 (17.4) | 22 (18.5) | 20 (31.3) | 8 (32.0) | 1 (4.0) | 10 (52.6) | 5 (29.4) | 4 (23.5) | 2 (22.2) | 4 (36.4) |
| Risk communication/ advice | 2 (1.7) | 14 (7.6) | 1 (0.7) | - | - | - | 3 (12.0) | - | - | - | - | - |
| Referral | 1 (0.9) | 3 (1.6) | 2 (1.3) | - | - | - | - | - | - | 2 (11.8) | 1 (11.1) | - |
| **Outcomes** | | | | | | | | | | | | |
| Attainment of risk factor targets | 4 (3.4) | 30 (16.3) | 43 (28.9) | 18 (15.1) | 2 (3.1) | 2 (8.0) | 1 (4.0) | 2 (10.5) | 6 (35.3) | - | 1 (11.1) | - |

"-" denotes indicators not reported;

[a]General population with no existing risk factors;

[b]At risk of CVD categorised from Framingham risk equation or World Health Organization (WHO)/ International Society of Hypertension (ISH) risk scores or high risk identified based on the presence of at least three of the following risk factors: age (males $\geq$ 45, females $\geq$ 55), smoker, hypertension, and dyslipidaemia;

[c]Others includes coronary artery disease, peptic ulcer, periphery artery disease, obesity, chronic obstructive pulmonary disease, and rheumatic heart disease; TIA, transient ischaemic attack; CVD, cardiovascular disease;

[d]Some of the organisation of care indicators might be relevant to other disease conditions as well.

articles [21, 37, 38, 47, 50, 52, 59, 72, 86, 88], glucose-lowering medications (13/726 indicators) reported in 10 articles [13, 21, 44, 48, 60, 62, 63, 77, 84, 86], prescription of aspirin (10/726 indicators) reported in six articles [50, 54, 59, 71, 74, 80]. The least reported indicators were on smoking cessation medications (8/726 indicators) reported in five articles [48, 50, 60, 62, 87], antiplatelet medications (3/726 indicators) in three articles [50, 72, 77]. Adherence or compliance to medications was reported in two articles [53, 81]. Appropriateness of medication prescribed reported in a single article [37], and management of depression or other mental illnesses was also reported in a single article [63].

*Risk communication or advice.* Quality indicators on risk communication or advice were reported only in seven articles (12.3%), contributing to 20/726; 2.8% of all indicators. Most commonly reported indicators were on care planning (5/726 indicators) in five articles [13, 45, 46, 50, 68]. Least reported indicators were instructions or pharmaceutical opinion for blood

pressure control, lipid lowering, or glucose monitoring [44], communication with patients [20], and information dissemination using leaflet reported in a single article [46]. There were no indicators focusing on risk communication or advice specifically for individuals with hypertension, atrial fibrillation, kidney disease, dyslipidaemia, transient ischemic attack, or smoking habits.

*Referral indicators*. Quality indicators for referral to other programs, specialist or further investigation were included in six articles (10.5%). Overall there were 1.2% of referral indicators (9/726) that included quality indicators that were developed to assess referrals to specialist or other programs such as smoking cessation programs were reported in two articles [60, 62], and referral to dietitians or weight loss programs reported in two articles [60, 62]. Referral indicators also included further investigations such as computed tomography (CT) scans or any other referrals reported in a single article [70]. Similarly, only one article each reported indicators on referral for physical activity [50], and an education program for diabetic patients [47]. There were no referral indicators for patients with hypertension, atrial fibrillation, kidney disease, mental health conditions, or dyslipidaemia.

**Outcome indicators.** Outcome indicators assess the achievement of risk factor targets as per the country specific clinical guidelines. Forty articles (70.2%) reported outcome indicators comprising 14.9% (108/726) of indicators. Outcome indicators included the attainment of targets related to blood pressure (43/726 indicators) reported in 28 articles [21, 41, 42, 44–47, 51, 53–55, 59, 61, 63–66, 70, 71, 74, 75, 80–84, 87, 89], attainment of serum lipids targets (32/726 indicators) reported in 24 articles [13, 20, 21, 42, 44–47, 53, 55, 56, 59, 61, 63, 65, 69–71, 75, 79, 82, 87, 89, 90], attainment of blood glucose targets (21/726 indicators) reported in 15 articles [21, 42, 44, 45, 47, 55, 61, 63, 65, 68, 74, 82, 85, 87, 89], attainment of multiple risk factors (8/726 indicators) were reported in five articles [59, 68, 76–78]. Outcome indicators for the attainment of the lifestyle advice were not reported.

## Methodological quality of indicators reported in the articles (including grey literature)

An evaluation of the articles using AIRE produced scores ranging from 31% to 100%. Only 26 articles (413/726 indicators) scored 50% or higher across all four domains (Fig 3). For AIRE domain 1: Purpose, relevance and organisational context (with a mean score of 77.2%), majority of the articles had sufficient information on the purpose of indicator, criteria for selection, organisational context for the development of indicator, and details on health care process. However, seven articles (12.3%) lacked details for this domain. For AIRE domain 2: Stakeholder involvement (mean score: 52.8%), 27 articles (47.3%) had missing information regarding the engagement of relevant stakeholders for developing the indicator and the formal endorsement process. For AIRE domain 3: Scientific evidence (mean score: 48.4%), 26 articles (45.6%) lacked information on the systematic methods for development of indicator and details on supporting evidence for critical appraisal. For AIRE domain 4: Additional evidence, formulation, usage (mean score: 61.6%), 13 articles (22.8%) had no frequency of collection, disaggregation details, and missing information regarding the validity, reliability, discriminative power, and pilot testing of these quality indicators. About 35.4% (257/726) indicators had included information about the frequency of collection of quality indicators as a part of their definitions. There were variations in recording and reporting of age categories; approximately 66.0% (479/726) of these indicators had a defined age category as a part of its definition.

Of the articles with strong methodological quality (26/57), there were 413/726 quality indicators. Three out of four were process indicators, followed by structural indicators (13.6%)

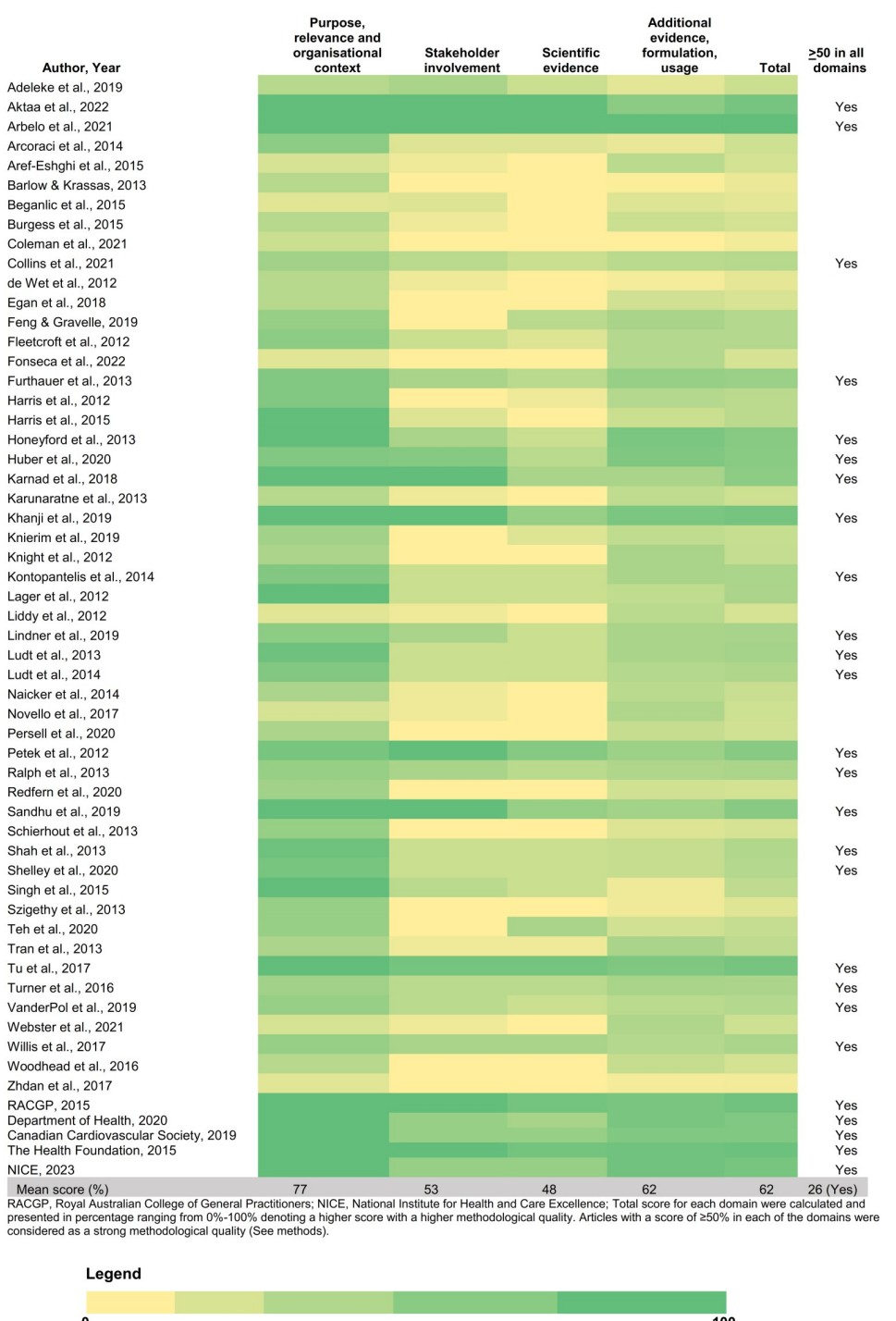

**Fig 3. Heat map presenting appraisal of articles by AIRE instrument and score by domains.**

and outcome indicators for the attainment of risk factor target (12.1%). Least reported (0.7%) was referral indicators. Majority of strong quality indicators were developed for individuals with a high risk of CVD (n = 78), followed by those with no existing risk factors (n = 73), diabetes (n = 71), atrial fibrillation (n = 56), and hypertension (n = 52). Quality indicator

developed for the other conditions such as kidney disease or dyslipidaemia lacked details on different domains of AIRE instrument.

## Discussion

To the best of our knowledge, this systematic review is the first to identify, summarise and appraise methodological quality of indicators reported in the articles for primary prevention of CVD in primary care. In this systematic review, 57 articles (quality indicator sets) with 726 quality indicators across 50 themes were included. Of the total indicators, 681 (93.8%) were unique indicators to specific risk factors or disease conditions. It is important to have indicators across all quality framework categories of primary care for comprehensive CVD prevention. The identified themes and care domains of these quality indicators for primary care, particularly those with strong methodological quality, would enable selection of minimum standard set of indicators to routinely monitor for the primary prevention of CVD.

The identified quality indicators were primarily focussed on the assessment of risk factors and prescription of medications, with few addressing adherence to medications, referral processes by disease conditions (i.e., referral for services for those with mental health conditions) or organisation of care. Although lifestyle risk factors are being assessed in primary care, quality indicators related to advice on healthy lifestyle were largely absent. Lifestyle advice is a critical element of care for the primary prevention of CVD and inadequate indicators could lead to low monitoring and follow-up [91]. In a study with American adults, authors reported that lifestyle advice from their general practitioners led to behavioural changes, such as reduction in weight or increased physical activity, thus, improving primary prevention of CVD [92]. This low reporting of lifestyle advice could be potentially due to incomplete records of assessment or advice on lifestyle risk factors [93–95]. Since more than 80% of risk factors are modifiable [96, 97], the inclusion of lifestyle management quality indicators, particularly advice on lifestyle risk factors, is essential. Despite the recommendations in existing primary prevention of CVD guidelines [4, 98], and stress being a significant risk factor, we found a notable lack of indicators focusing on psychosocial factors such as depression, stress or mental health illness. Another critical area identified was the need for development of indicators related to environmental exposures, such as indoor or outdoor pollution, recognising these as emerging risk factors for CVD in primary care. There is also growing evidence on the effectiveness of immunisation or vaccination as part of strategies relevant to the primary prevention of CVD [99], including uptake or counselling on vaccine effectiveness [100]. Therefore, quality indicators related to assessment of environmental exposure (i.e., indoor or outdoor pollution), advice on healthy lifestyle by disease conditions, and assessment or advice on vaccination needs to be developed or integrated into the monitoring for primary prevention of CVD.

There was a limited number of outcome indicators developed for evaluating the attainment of individual risk factor targets. This is consistent with the previous review on quality indicators for acute CVD care [101]. Overall, reasons for the small number of outcome indicators may be the length of time it takes to assess these outcomes, implied by documentation (thresholds met or not) or the availability of data for reporting. Although the primary care providers might need detailed information about the process of CVD care for quality improvement or research purposes; funders, program planners and the public might be more interested in structure or outcomes of care. Structure indicators (e.g., patient perspective, capacity building) has an impact on process (e.g., assessment of blood pressure) or outcome indicators (e.g., attainment of blood pressure targets) as a part of primary prevention of CVD. Outcome indicators are also important to enable monitoring the overall result of the course of action or treatments provided, than a single aspect of care [17]. However, monitoring of the structure

indicators and its outcome is often neglected in primary care [102]. This could be because structural interventions are considered less important for the primary prevention of CVD (e.g. stroke) [103]. A combination and balance of structure, process and outcome quality indicators aligned with the Donabedian framework is important for measuring the quality of care for CVD prevention in primary care settings [24]. The inadequacy of structure and outcome indicators can impede CVD and stroke prevention measures as this may lead to inadequate monitoring of the patient outcomes, infrastructure support and accountability [104]. These indicators are critical to deliver high- quality care especially targeting monitoring and following up those with sub-optimal outcomes [105]. There is a need for the right mix of the indicators, including structure and outcome indicators, to identify the gaps in workforce or infrastructure, ensure quality for primary prevention, and monitor the patient outcomes [106]. Importantly, short term (repeated measures of the lifestyle risk factors), intermediate outcome indicators such as control of risk factors (e.g., attaining blood pressure, blood glucose, smoking cessation targets) are needed for each risk factors for CVD prevention in primary care settings.

In this review, we summarised the available quality indicators used globally for comprehensive inclusion of all indicators and appraised their methodological quality. Consistent with other studies [32, 34, 35], the methodological quality appraisal demonstrated that most of the quality indicators included in the articles lacked rigour in the reporting of the development process. Fundamental development processes such as the evidence generation, stakeholder's engagement, and pilot testing of the quality indicators were missing in many articles. Of those articles with strong methodological quality, they were developed either through literature review followed by the Delphi technique or as part of quality improvement initiatives. Quality indicators extracted from implementation or evaluation articles were lacking with the detailed information on age categories, target population, reporting frequencies, numerators and denominators, which varied substantially across the articles. This comprehensive summary will enable countries to choose the feasible indicators depending upon the clinical guidelines, socio-economic context, health system financing, or cultural context specifically in low-and middle-income countries with limited evidence on quality indicators.

Although the evidence-based guidelines and recommendations for the management of risk factors are similar across countries, clinical practice is context dependent and will vary between countries. This review is a first step for development of quality indicators, highlighting the distribution of quality indicators worldwide, emphasising the necessity for countries to tailor their quality indicators to monitor their specific context and the burden of CVD. There is already a recommended set of quality indicators to improve quality of care in patients with atrial fibrillation [37]. However, there is need for developing such specific quality indicators for individuals with pre-existing conditions (e.g., kidney disease, dyslipidaemia) to ensure their ongoing assessment and optimal care during each visit.

Despite the implementation of incentive schemes aimed at enhancing the recording and reporting of quality indicators, their effectiveness has been limited in some countries [39–43]. Therefore, it is imperative to explore and identify innovative initiatives to improve the monitoring of CVD prevention in primary care settings. To enhance the collection of quality indicators, adopting local, bottom-up planning (termed as "microplanning") which has been evidenced to be successful in other healthcare programs [107, 108], could potentially be a step forward for CVD prevention in primary care settings. For example, this approach would facilitate the routine assessment and recording of risk factors during every follow-up visit, fostering tailored advice for lifestyle modifications.

### Strengths and limitations

The strengths of this review lie in its rigorous methodology, including a registered protocol in PROSPERO and engagement of an advisory group comprising topic experts. Involving three team members in initial title or abstract and full-text screening minimised the risk of overlooking relevant information. A significant contribution of this review was to identify and summarise quality indicators for the prevention of CVD in primary care, offers valuable guidance for monitoring and tracking primary care services. However, there are several limitations to consider. Firstly, definition of quality indicators and frequency of collection varied considerably, limiting applicability of quality indicators across all primary care settings. Secondly, our restriction to articles in the English language may have resulted in the omission of important quality indicators, particularly from non-English-speaking settings. This might have partly led to fewer articles being identified (n = 2) from low- and middle-income countries [54, 67], limiting generalisability. Thirdly, the quality appraisal of the quality indicators, may be constrained by the limited availability of information reported on the development of these quality indicators described in the included articles. Lastly, quality indicators were often developed to align with country specific practice guidelines, whereby mapping back to the clinical guideline recommendations were not undertaken as part of this review.

### Conclusion

The adoption of continuous quality assessment and improvement initiatives, including indicators for the primary prevention of CVD, has the potential to improve the quality of care provided. Given the worldwide growing burden of CVD, the appraisal of quality indicators can serve as a starting point for monitoring various domains of primary care for CVD prevention. We identified many processes of care indicators, but there was lack of structure and outcome indicators, thereby limiting the full scope of quality assessment. The findings from this review will support patients, clinicians, healthcare organisations and policymakers in selecting relevant quality indicators to expand indicator development to monitor the quality of care. In the future, data collection tools might need to be developed that permit standardised collection and between country comparisons. To achieve this, consensus for numerators, denominators (excluded populations), the period of assessment, sources of data, rationale for indicators aligned with clinical recommendations, and methods of reporting or challenges in implementation should be defined.

### Supporting information

**S1 Table. Overview of search strategy.**
(DOCX)

**S2 Table. Overview of grey literature search strategy: List of organisations and webpage reviewed.**
(DOCX)

**S3 Table. Full-text screening checklist.**
(DOCX)

**S4 Table. Study characteristics extraction table template.**
(DOCX)

**S5 Table. Quality indicators data extraction table template.**
(DOCX)

**S6 Table. Appraisal of Indicators through Research and Evaluation (AIRE) instrument criteria to evaluate quality indicators.**
(DOCX)

**S7 Table. Articles identified in database search and reasons for exclusion.**
(XLSX)

**S8 Table. Data extraction and synthesis sheet.**
(XLSX)

**S9 Table. List of unique quality indicators by conditions.**
(DOCX)

**S10 Table. Study characteristics extraction sheet.**
(DOCX)

**S11 Table. Summary of quality indicators reported in the articles as per Donabedian framework.**
(DOCX)

**S1 File. PRISMA 2020 checklist.**
(DOCX)

## Acknowledgments

We would like to thank the research librarian from Monash University for the support in finalising the search strategy. We also express our gratitude to the members of the Stroke and Ageing Research Group for their support during the preparation of this manuscript. Special thanks to Dr Rajshree Thapa for her review and edits in the manuscript.

## Author Contributions

**Conceptualization:** Kiran Bam, Monique F. Kilkenny.

**Formal analysis:** Kiran Bam, Monique F. Kilkenny.

**Methodology:** Kiran Bam, Muideen T. Olaiya, Dominique A. Cadilhac, Monique F. Kilkenny.

**Project administration:** Kiran Bam, Dominique A. Cadilhac, Monique F. Kilkenny.

**Resources:** Monique F. Kilkenny.

**Supervision:** Muideen T. Olaiya, Dominique A. Cadilhac, Monique F. Kilkenny.

**Validation:** Dominique A. Cadilhac.

**Visualization:** Kiran Bam.

**Writing – original draft:** Kiran Bam.

**Writing – review & editing:** Muideen T. Olaiya, Dominique A. Cadilhac, Julie Redfern, Mark R. Nelson, Lauren M. Sanders, Vijaya Sundararajan, Nadine E. Andrew, Lisa Murphy, Monique F. Kilkenny.

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
