## [Decision Letter · Decision Letter 0]

23 Jun 2024

PONE-D-24-16110Quality indicators for the primary prevention of cardiovascular disease in primary care: A systematic reviewPLOS ONE

Dear Dr. Kilkenny,

Thank you for submitting your manuscript to PLOS ONE. After careful consideration, we feel that it has merit but does not fully meet PLOS ONE’s publication criteria as it currently stands. Therefore, we invite you to submit a revised version of the manuscript that addresses the points raised during the review

We look forward to receiving your revised manuscript.

Kind regards,

André Ramalho, PhD

Academic Editor

PLOS ONE

[DAC received restricted grants from Boehringer Ingelheim, Bristol-Myers, Moleac and Medtronic; and DAC and MFK from Amgen, outside the submitted work.]. 

Please include your updated Competing Interests statement in your cover letter; we will change the online submission form on your behalf."

4. Please remove your figures from within your manuscript file, leaving only the individual TIFF/EPS image files, uploaded separately. These will be automatically included in the reviewers’ PDF.

Additional Editor Comments (if provided):

Reviewers' comments:

Reviewer's Responses to Questions

**Comments to the Author**

1. Is the manuscript technically sound, and do the data support the conclusions?

Reviewer #1: Partly

Reviewer #2: Yes

2. Has the statistical analysis been performed appropriately and rigorously? 

Reviewer #1: Yes

Reviewer #2: Yes

3. Have the authors made all data underlying the findings in their manuscript fully available?

Reviewer #1: Yes

Reviewer #2: Yes

4. Is the manuscript presented in an intelligible fashion and written in standard English?

Reviewer #1: Yes

Reviewer #2: Yes

5. Review Comments to the Author

Reviewer #1: Thank you for this study. The public health case for this study is clearly necessary.

How was the screening done? It may be useful to include information on the process of screening the articles. Typically, the articles will first be screened by title, then abstract, and finally full-text screening (this information was sort of provided under the strengths and limitations, but it would be helpful to clarify this under methods). The authors did a good job by providing the respective role of the reviewers, but they stop short of detailing the screening process. Also, were any of the articles eliminated using automated tools?

Authors should justify the tools, particularly the critical appraisal tool and framework for assessing the quality indicators that were used.

Authors should consider providing a sample of the search terms, preferably as a search string. This bolsters the methodological integrity of the study and promotes its reproducibility.

Given that the concept of the review is clinical in nature, the authors may want to check the search terms in MeSH to ensure that they captured all the possible variations of the search terms.

The authors indicated that they used the PRISMA checklist for systematic reviews, but the following aspects of the checklist were inadequately tackled: eligibility criteria (item #5); search strategy (item #7); and synthesis methods (item #13).

How did the authors find the grey literature? It appears that such grey literature were pulled from sources that were outside the four stated databases. Please clarify.

The rationale of the study is towards stroke prevention, however, the authors barely touched on this during the discussion. The authors should consider discussing the implications of the findings, particularly the finding that most of the indicators focused on assessment risk factors at the expense of structure and outcome indicators, on stroke prevention. Could it be that the inadequacy of structure and outcome indicators undercut the stroke prevention measures? If so, how?

Reviewer #2: i accept manuscript which discuss the quality indicators for primary CVD prevention Quality indicators for the primary prevention of cardiovascular disease in primary care A systematic review for publication

6. PLOS authors have the option to publish the peer review history of their article (what does this mean?). If published, this will include your full peer review and any attached files.

Reviewer #1: No

Reviewer #2: No

---

## [Author Response · Author response to Decision Letter 0]

28 Jul 2024

Please see response to reviewer's document

---

## [Decision Letter · Decision Letter 1]

2 Oct 2024

Quality indicators for the primary prevention of cardiovascular disease in primary care: A systematic review

PONE-D-24-16110R1

Dear Dr. Kilkenny,

We’re pleased to inform you that your manuscript has been judged scientifically suitable for publication and will be formally accepted for publication once it meets all outstanding technical requirements.

Kind regards,

André Ramalho, PhD

Academic Editor

PLOS ONE

Reviewers' comments:

Reviewer's Responses to Questions

**Comments to the Author**

1. If the authors have adequately addressed your comments raised in a previous round of review and you feel that this manuscript is now acceptable for publication, you may indicate that here to bypass the “Comments to the Author” section, enter your conflict of interest statement in the “Confidential to Editor” section, and submit your "Accept" recommendation.

Reviewer #3: (No Response)

Reviewer #4: All comments have been addressed

2. Is the manuscript technically sound, and do the data support the conclusions?

Reviewer #3: Yes

Reviewer #4: Yes

3. Has the statistical analysis been performed appropriately and rigorously? 

Reviewer #3: Yes

Reviewer #4: N/A

4. Have the authors made all data underlying the findings in their manuscript fully available?

Reviewer #3: Yes

Reviewer #4: Yes

5. Is the manuscript presented in an intelligible fashion and written in standard English?

Reviewer #3: Yes

Reviewer #4: Yes

6. Review Comments to the Author

Reviewer #3: 1. Introduction: The authors might consider a small comment about primary care accessibility - this might have a significant impact in the success of preventive strategies.

2. Methods/Eligibility criteria: This article focuses on finding and categorizing quality indicators concerning preventive care to avoid CVD disease. As such, I would suggest approaching the research as a scoping review and forfeiting the use of a PICO question. If the authors do want to use a PICO question, then it should be reformulated to something like:

P - patients at risk for the development of CVD disease

I - use of an ideal set of primary care indicators for prevention strategies

C - actual standard of care

O - development of CVD disease

Still, this would be a good investigation question for a different work - one in which the authors could study the impact of the application of quality indicators in primary care services.

3. Results: Considering how many indicators the authors found concerning specific areas (i.e. 55 indicators concerning blood pressure monitoring, 36 indicators about smoking status), I would advise for an exclusion of duplicates or similar indicators. All the data concerning the indicators in the results reports to the total of indicators found (726), and not the 681 unique indicators.

4. Discussion: The fact that many of these indicators are used in pay for performance systems may also account for the fact that most of them are related to process, rather than structure or outcome.

Thank you for this research paper. I would recommend checking "BILHETE DE IDENTIDADE DOS INDICADORES DE

CONTRATUALIZAÇÃO DOS CUIDADOS DE SAÚDE PRIMÁRIOS", a document created by the portuguese healthcare central administration defining a set of primary care indicators and respective definitions, some of which concerning CVD disease and prevention. The 2016 version can be found on https://www2.acss.min-saude.pt/Portals/0/bilhete_identidade_indicadores_contratualizacao_2016___2016_02_16.pdf

Reviewer #4: I read with interest the article by Bam et al. and reviewed the previous comments from the reviewers. This is a comprehensive study, and I appreciate the authors' efforts, which are well-presented. Overall, I believe the manuscript is well-prepared. The authors might consider moving the search strategy from the main text to the supplementary file to improve the readability of the manuscript.

7. PLOS authors have the option to publish the peer review history of their article (what does this mean?). If published, this will include your full peer review and any attached files.

Reviewer #3: **Yes: **Pedro Manuel Correia Castro

Reviewer #4: No

---

## [Editor Report · Acceptance letter]

8 Oct 2024

PONE-D-24-16110R1 

PLOS ONE

Dear Dr. Kilkenny, 

I'm pleased to inform you that your manuscript has been deemed suitable for publication in PLOS ONE. Congratulations! Your manuscript is now being handed over to our production team.

Kind regards, 

on behalf of

Prof. Dr. André Ramalho 

Academic Editor

PLOS ONE